# P14^ARF^: The Absence that Makes the Difference

**DOI:** 10.3390/genes11070824

**Published:** 2020-07-20

**Authors:** Danilo Cilluffo, Viviana Barra, Aldo Di Leonardo

**Affiliations:** 1Department of Biological, Chemical and Pharmaceutical Sciences and Technologies, University of Palermo, 90128 Palermo, Italy; danilo.cilluffo@unipa.it (D.C.); viviana.barra@unipa.it (V.B.); 2Centro di Oncobiologia Sperimentale (C.O.B.S.) via San Lorenzo, 90146 Palermo, Italy

**Keywords:** p14^ARF^, aneuploidy, CENP-E, GSK923295

## Abstract

P14^ARF^ is a tumor suppressor encoded by the CDKN2a locus that is frequently inactivated in human tumors. P14^ARF^ protein quenches oncogene stimuli by inhibiting cell cycle progression and inducing apoptosis. P14^ARF^ functions can be played through interactions with several proteins. However, the majority of its activities are notoriously mediated by the p53 protein. Interestingly, recent studies suggest a new role of p14^ARF^ in the maintenance of chromosome stability. Here, we deepened this new facet of p14^ARF^ which we believe is relevant to its tumor suppressive role in the cell. To this aim, we generated a monoclonal HCT116 cell line expressing the p14^ARF^ cDNA cloned in the piggyback vector and then induced aneuploidy by treating HCT116 cells with the CENP-E inhibitor GSK923295. P14^ARF^ ectopic re-expression restored the near-diploid phenotype of HCT116 cells, confirming that p14^ARF^ counteracts aneuploid cell generation/proliferation.

The alternate open reading frame (ARF) gene was originally identified as an alternative transcript of the CDKN2a locus on human chromosome 9p21 that encodes the nucleolar protein p14^ARF^ which has a potent tumor suppressor activity. Loss of the CDKN2a locus is the second most frequent copy number alteration that characterizes human tumors and cancer cell lines. One of the most well-defined functions of p14^ARF^ protein is to suppress aberrant cell proliferation in response to oncogene insults by activating the transcription factor p53 that, in turn, triggers the expression of many apoptosis inducers and cell cycle inhibitory genes [1]. In the presence of oncogenic stimuli p14^ARF^ binds the p53-specific E3 ubiquitin ligase Mouse double minute 2 homolog (Mdm2) and keeps it in the nucleolus. This event prevents the interaction between Mdm2 and p53 by blocking the transport to the cytoplasm of p53 and its degradation by the proteasome Mdm2-mediated [1,2,3]. ARF is also able to inhibit the ubiquitin ligase activity of ARF-BP1/Mule protein. It has been reported that in p53 wild-type cells ARF-BP1 directly binds and ubiquitinates p53, and inactivation of endogenous ARF-BP1 is crucial for ARF-mediated p53 stabilization [4]. 

On the other hand, there is some evidence that ARF, especially its small mitochondrial isoform (smARF) [5], also has the ability to restrain cell growth independently of the presence of the p53 protein. The Zambetti group showed that triple knockout mice lacking *ARF*, *p53*, and *MDM2* genes were more tumour prone when compared to mice lacking only *p53* and *MDM2* [6]. In addition and in line with this result they showed that ARF overexpression can induce the arrest in the G1 phase of the cell cycle in cells lacking p53 [6]. Moreover, ARF through interaction with a multitude of different target proteins (i.e., NPM1, E2F, Myc) regulates mitochondrial activity, autophagy, sumoylation, transcription activities, specific protein turnover, and ribosome biogenesis, leading to cell cycle arrest and senescence, as reviewed by Ko and colleagues [7]. The importance of the role of p14^ARF^ is highlighted by the effect of its loss in mouse models: mice were highly prone to develop tumors and mostly died during their first year of life. In addition, p14^ARF^-null mouse embryonic fibroblasts (MEFs) can overcome the senescence crisis and proliferate continuously, and they also can be easily transformed by the Ras oncogene [8]. Considering that p14^ARF^ is a tumor suppressor it is not surprising that most human cancers (colorectal, gastric, prostate, breast bladder, oral, hepatocellular, lung, leukaemia, glioblastoma, and melanoma) are characterized by its inactivation mainly due to its promoter hypermethylation, homozygous deletions and mutations (reviewed in [1]) or just reduced levels whose cause has not been elucidated yet [9]. Due to its frequent alteration in cancer, population studies have been published to show with the most recent genome-wide approaches, namely next generation sequencing and array-CGH, the most recurrent p14ARF alterations in melanoma and leukaemia [10,11]. ARF protein undoubtedly is a critical player in responding to an oncogenic insult, like hyperproliferative signal, however its role in another cancer-related aberration, aneuploidy (loss or gain of entire chromosomes), has not been punctually explored so far. Intriguingly, in the last decade experimental evidences of this role of p14^ARF^ or its murine orthologue p19^ARF^ have been published. In fact, Weaver group showed that ARF loss induced a small degree of aneuploidy in murine embryonic fibroblasts (MEFs) that was the result of chromosome mis-segregation [12,13]. They also observed an increase of the aneuploidy level when CENP-E loss was coupled with ARF loss [12]. On the other hand, we constantly observed that p14^ARF^ is involved in the cell response to the aneuploid phenotype triggered by the impairment of different types of proteins, namely MAD2 and DNMT1 [14,15]. In particular, we showed that p14^ARF^ can respond either by arresting the cell cycle and entering senescence if necessary or by inducing apoptosis [14,15,16]. Altogether, these results would let us think that the presence of p14^ARF^ is critical to halt aneuploidy. To this regard, we recently demonstrated that when CENP-E is reduced by 50% and aneuploidy is consequently induced, HCT116 cells, where p14^ARF^ is not expressed [17], are not able to recover their euploidy. Conversely, normal IMR90 human fibroblasts, in which aneuploidy is induced, restore their diploid chromosomal content in four weeks accompanied by a strong p14^ARF^ induction. HCT116 cells could finally recapitulate IMR90 cell behaviour and recover from aneuploidy once they ectopically re-express p14^ARF^ [18]. These evidences demonstrate that p14^ARF^ is involved in blocking the proliferation of aneuploid cells. However, we only analysed the effects of p14^ARF^ ectopic expression in a heterogenous population of HCT116 cells expressing different levels of p14^ARF^ and we did not investigate on the mechanism that allows the elimination of the aneuploid cells.

Here, we generated a monoclonal HCT116 cell line harbouring p14^ARF^ cDNA integrated in the genome specifically at ‘TTAA’ sites by the PiggyBac transposon vector system (ePB) (Figure 1A,B). Briefly, HCT116 cells were co-transfected with the ePB vector carrying the p14^ARF^ c-DNA and the vector carrying the transposase enzyme (hyPB). Transfected cells were then selected with the antibiotic blasticidin for ten days and then single colonies were picked, analysed and one positive clone was chosen for further investigation. In the engineered HCT116 cells (HCT116 ePB cells), p14^ARF^ cDNA expression is under the control of the TET-On technology that allows activation of gene expression by administration of doxycycline. By using a low dose of doxycycline (10ng/mL) we obtained the consistent expression of p14^ARF^ (Figure 1C–E). Reverse transcription PCR (RT-qPCR) analysis showed that after doxycycline treatment the expression levels of p14^ARF^ are higher in HCT116 ePB-p14ARF cells in comparison to control HCT116 cells (Figure 1C). Moreover, immunofluorescence and Western blot analyses showed the presence of the ectopic p14^ARF^ protein (Figure 1D,E). Following p14^ARF^ ectopic re-expression we assessed whether in our model system cell proliferation and normal cell cycle progression was affected. To this aim, we measured cell proliferation rates and analysed cell cycle by cytofluorimetry. These experiments showed that p14^ARF^ ectopic re-expression does not affect either cell growth or cell cycle progression (Figure 1F,G). 

We then induced aneuploidy in these engineered HCT116 cells by treating them for 72 h with the CENP-E inhibitor GSK923295 [19]. GSK923295 is an allosteric inhibitor, which specifically targets the ATPase binding site of the CENP-E’s motor domain and non-competitively inhibits the ATPase activity of CENP-E with high selectivity [20] but does not affect the protein stability, thus avoiding any unwanted effect due to kinetochore architecture alteration. We treated the cells with a low dose (5nM) of GSK923295 to allow a partial though efficient CENP-E inhibition without the cytotoxic effect typical of this drug [18], to investigate the fate of the aneuploidy daughter cells following the p14^ARF^ induction/re-expression (Figure 2A,B). As expected, following CENP-E inhibition 65% of HCT116, ePB-p14 cells become hypodiploid (compared to 18,5% in control cells (DMSO)). After doxycycline treatment, and consequent p14^ARF^ expression, the percentage of hypodiploid cells dropped down to 24% that is comparable to control cells. We had previously observed a similar behaviour in a population of HCT116 cells harbouring exogenous p14^ARF^. In these cells, p14^ARF^ was under the control of a Tet-off system, therefore only when doxycycline was absent could p14^ARF^ be expressed. We showed that when CENP-E was reduced by RNA interference and there was no p14^ARF^ expression (upon doxycycline treatment), these cells became aneuploid (50%). However, when doxycycline was washed out and thus p14^ARF^ was expressed, the percentage of aneuploid cells was reduced from 50% to 22% [18]. In addition, we also observed that, even if transiently transfected with a pcDNA3.1 expression vector, p14^ARF^ could partially reduce aneuploidy of CENP-E-depleted HCT116 cells from 73% to 56% [18]. Therefore, our new observations with a more accurate cell model system are perfectly in line with our previous results showing that p14^ARF^ ectopic re-expression restore the near-diploid phenotype of HCT116 cells. These results confirm that p14^ARF^ counteracts aneuploid cell generation/proliferation. It is well established that p14^ARF^ regulates cell cycle arrest and apoptosis, however it remains unclear if p14^ARF^ uses one or both mechanisms to fight aneuploidy [13,14,21]. To shed light on this we evaluated cell cycle progression and apoptosis of HCT116 cells expressing p14^ARF^ after treatment with GSK923295 (Figure 2C–F). Propidium Iodide staining and the cytofluorimetric analysis clearly showed that cells did not arrest their cell cycle when aneuploidy was induced by the treatment with the CENP-E inhibitor GSK923295 and p14^ARF^ was expressed. This made us think that apoptosis could likely be the way through which p14^ARF^ induces the clearance of the aneuploid cells. We hypothesized thus that this could be the mechanism through which p14^ARF^ helps the maintenance of chromosomal stability. Hence, to identify the presence of apoptotic cells, we performed acridine orange and ethidium bromide staining which discriminates live, apoptotic, and necrotic cells. By this mean we observed 25% of apoptotic cells in HCT116 cells expressing p14^ARF^ following GSK923295 treatment. These results strongly suggest that p14^ARF^ triggers the apoptotic pathway to halt aneuploid cell proliferation.

Through this mechanism numerical chromosomal stability would be preserved in normal cells. This is well-known during development of Drosophila melanogaster that indeed uses apoptosis to eliminate aneuploid cells [22]. In addition, it has also been observed that, in mouse models of pre-implantation chromosome mosaicism, the aneuploid cells were depleted from the embryo. This depletion is due by a lineage-specific apoptosis during blastocyst [23]. Chromosome instability (CIN) by itself is unfavourable for cell proliferation and thus detrimental. However, CIN, though it is not sufficient, alone, for cell transformation, can facilitate cancer initiation and progression in particular genetic backgrounds. Therefore, it is conceivable that adult cells can use apoptosis to counteract aneuploidy similarly as cells do during development. Nevertheless, to our knowledge up to now there is no animal model showing this mechanism in adult cells. Similarly, we are not aware of studies specifically describing the fate of cells with chromosome instability in patients that have p14^ARF^ mutations. Though, p14^ARF^ role in cancer as tumor suppressor is vastly demonstrated. Noteworthily, it has been noticed that patients with mutations in p14^ARF^ have high risk of developing melanoma and pancreatic cancer [24,25]. Furthermore, a thorough study of human cancers has highlighted that a large number of human tumors use several mechanisms (both genetic and epigenetic) to fully inactivate ARF [26]. Altogether, these evidences point out that p14^ARF^ has a fundamental role in fighting cell transformation and suggest that, due to the high frequency of deregulation in cancer, this role could be played at early stages of cell transformation. We think that this could be due to p14^ARF^ ability in counteracting aneuploidy. 

P14^ARF^ was firstly identified as an alternate transcript of the CDKN2a locus, which is the second most frequently mutated site in cancers after p53. Its tumor suppressor role as regulator of the cell cycle and apoptosis has been indeed well established. Classically, a hyperproliferative signal that results from an oncogenic insult triggers ARF activation. In this brief report, by confirming previous results, we suggest that p14^ARF^ can also respond to the onset of aneuploidy identification, thus providing an additional function as a sentry of the euploid set of chromosomes. Remarkably, aneuploidy, i.e., an abnormal number of chromosomes, is a hallmark of virtually all tumors. We observed that the absence of p14^ARF^ worsens the aneuploid phenotype of the cells, however it was generated. These cells, indeed, cannot either block their cell cycle or enter apoptosis, and therefore, as a matter of fact, the possibility to impede the proliferation of aneuploid cells is denied. The maintenance of aneuploidy can undoubtedly contribute to malignant transformation. The presence/restoration of p14^ARF^, instead, allows cells to counteract aneuploidy, most probably by sentencing the aneuploid cells to apoptosis. In fact, this mechanism of aneuploid cell elimination by apoptosis has been observed during development. It is conceivable that this function of ARF is an important part of its role as a tumor suppressor. Furthermore, it is generally established that gene copy number correlates with gene expression. It is not too daring to think that the imbalance of gene expression that is induced by aneuploidy could then generate a signal resembling the hyperproliferative stress typically sensed by p14^ARF^. This, in turn, could activate p53-dependent apoptosis. In conclusion, through this evidence, it is clear that p14^ARF^ not only responds to oncogene stimuli or to DNA damage, but also to an “aneuploidy signal” and protects the cells from the maintenance of such aberrant and potentially dangerous condition. 

## Figures and Tables

**Figure 1 genes-11-00824-f001:**
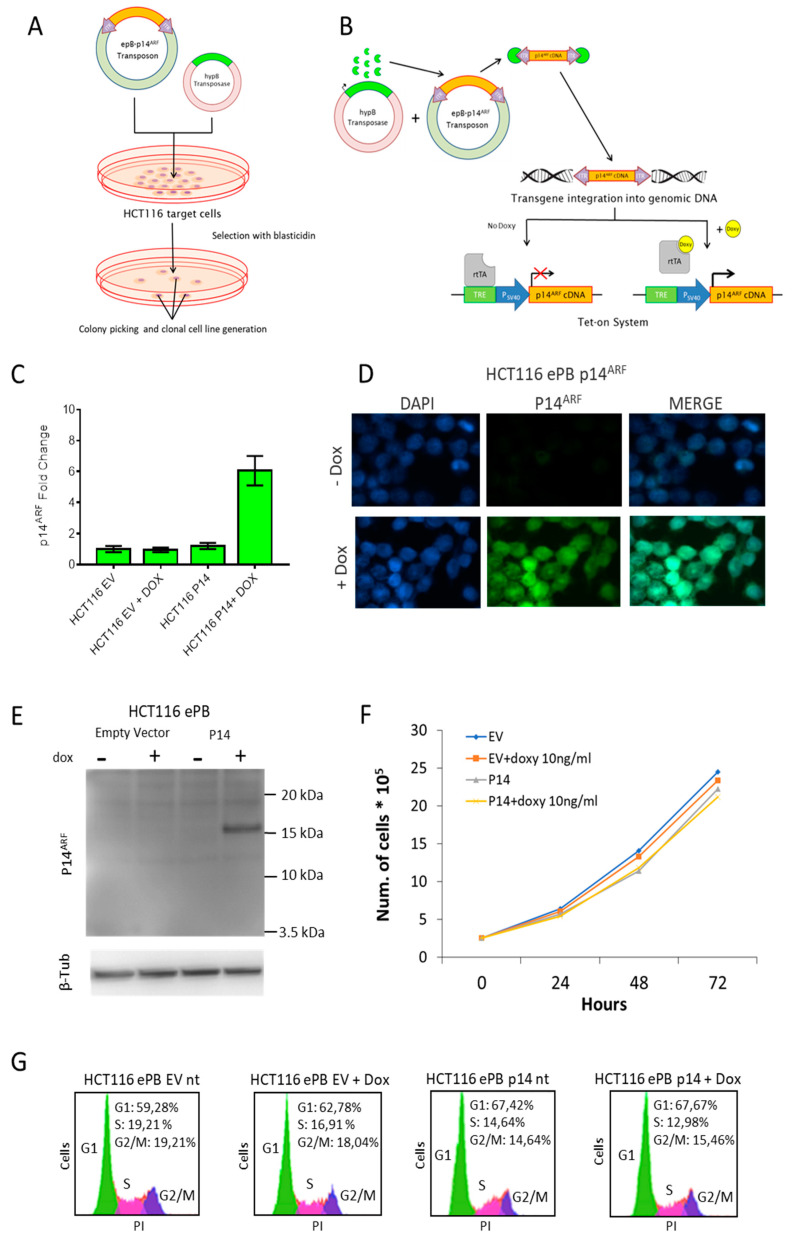
Generation and characterization of a monoclonal HCT116 cell line harbouring the p14^ARF^ cDNA. (**A**) Experimental schematics of the monoclonal cell line generation. (**B**) Outline of the PiggyBac transposon integration and Tet-on expression system; doxycycline (10ng/mL) was added to the cultured cells for 24 h to switch on p14^ARF^ expression. (**C**) RT-qPCR showing p14^ARF^ transcript levels in HCT116 transfected cells; GAPDH was used as calibrator for the normalization. Standard error of the mean is indicated. (**D**) Immunofluorescence and (**E**) western blot analyses confirmed p14^ARF^ protein increase in HCT116 ePB-p14^ARF^ 24h after doxycycline treatment. (**F**) Cell proliferation analysis of HCT116-ePB cells with and without doxycycline treatment. (**G**) Cytofluorimetric profiles of HCT116 ePB-p14^ARF^ cells. DNA was stained with Propidium Iodide (PI).

**Figure 2 genes-11-00824-f002:**
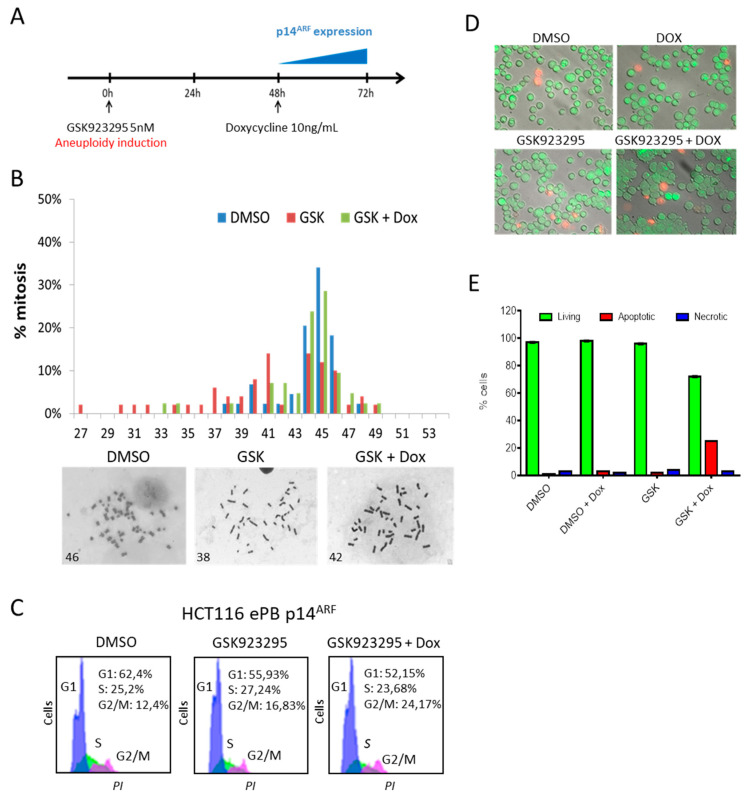
p14^ARF^ expression counteracts aneuploidy. (**A**) Schematics of the experiment, GSK923295 dissolved in DMSO (5nM) was given to the cells for 72 h. (**B**) Graphs and representative images showing the results of the metaphase spread assay of HCT116 ePB-p14^ARF^ cells (*n* = 30). (**C**) Cytofluorimetric profiles of HCT116 ePB-p14^ARF^ cells treated with GSK923295 and doxycycline. (**D**) Acridine orange and ethidium bromide (AO/EtBr) double staining of HCT116 ePB-p14^ARF^ cells. Representative pictures: acridine orange (AO) permeates all cells and makes green, ethidium bromide (EB) is only taken up by cells when cytoplasmic membrane integrity is lost and stains the nucleus of necrotic cells in red. (**E**) The graph summarizes the percentage of living, apoptotic and necrotic HCT116 ePB-p14^ARF^ cells. Standard error of the mean is indicated.

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
