# Peer review of "P14ARF: The Absence that Makes the Difference"

_genes, 2020, doi:10.3390/genes11070824_

Round 1

Reviewer 1 Report

In this potentially intriguing study, the authors suggest that ARF is functioning as a TSG independently of p53. The authors do not demonstrate any evidence to support this claim that ARF can mitigate aneuploidy without a mechanistic link to HCT116. What is the p53 status of HCT116? Does p53 expression change after ARF induction?

Author Response

Responses to Reviewers’ comments:

(in italics the Authors’ responses).

Reviewer 1

In this potentially intriguing study, the authors suggest that ARF is functioning as a TSG independently of p53. The authors do not demonstrate any evidence to support this claim that ARF can mitigate aneuploidy without a mechanistic link to HCT116. What is the p53 status of HCT116? Does p53 expression change after ARF induction?

We thank this Reviewer for the positive comment. However, we are puzzled by the conclusion s/he drew about our study. Here we have not mentioned that p14ARF counteracts aneuploidy independently from p53. Instead, we only cited in the introduction the existence of reports demonstrating that some p14ARF roles are p53 independent.

We have not analysed p53 involvement in this work, since we already demonstrated that p53 presence is necessary for p14ARF to eliminate aneuploid HCT116 cells, generated by MAD2 depletion, and that, when p14ARF is ectopically expressed in these cells, p53 is over-expressed (Veneziano et al. 2016, cited in the text).

In this manuscript, although we generated aneuploidy by a different strategy, there is no reason to think that the p14ARF response pathway is not working in the same way.

HCT116 are p53-competent cells, and our results suggest that p14ARF is the first line of defence against aneuploidy, it is the sentinel that warns and triggers the response likely p53-mediated. Without p14ARF, even if functional, p53 cells cannot control aneuploidy.

Anyway, we take note of this Reviewer’s comment and for the sake of clarity we have modified one sentence in the abstract which could have been misleading:

“Most of these activities are notoriously mediated by p53 but accumulating evidence draws attention to additional p53-indipendent functions played through interactions with other proteins.”

Changed in: “ P14ARF functions can be played through interactions with several proteins, however the majority of its activities are notoriously mediated by p53” .

Reviewer 2 Report

In this article, the authors reported that p14ARF contributes to chromosomal stability in the HCT116 cells. However, current results are still insufficient for the insists because of statistical and cell biological concerns.If some key concerns could be addressed, I think this would be suited for Genes otherwise the manuscript might be better submitted in a journal focused on the interests of oncology.

Major concerns:

  1. In this work, the authors used the single clone of HCT116 cells with tet-on system-mediated p14ARFexpression to performed all experiments. The clonal effect on these phenotypes should be evaluated.

  1. In Figure 2B, they scored the chromosomal instability using the criteria of <2n, 2n, and >2n. However, this representation is not general. They should show the number or percentage of aneuploid cells in each chromosome number. In general, aneuploidy followed by the spindle assembly checkpoint disruption accompanies both loss and gain of chromosomes. In this manuscript, they showed only hypo-diploidy after the CENP-E inhibitor treatment. To exclude the possibility of artifacts in the preparation of chromosome slides, images of chromosomes in each sample should be shown.

  1. HCT116 cells are immortalized cells. It is too difficult to analyze the cellular senescence in the HCT116 cells. Although SA-b-Gal is indeed a senescence marker, it is too risky to conclude the insists based on only Figure 2D. Thus, I recommend to delete the data and descriptions of cellular senescence.

Minor concerns

  1. The error bars of all graphs are standard deviations or standard errors? They should clarify this point.

  1. They did not perform any statistical analyses.

Author Response

Responses to Reviewers’ comments:

(in blue and italics the Authors’ responses).

Reviewer 2

In this article, the authors reported that p14ARF contributes to chromosomal stability in the HCT116 cells. However, current results are still insufficient for the insists because of statistical and cell biological concerns. If some key concerns could be addressed, I think this would be suited for Genes otherwise the manuscript might be better submitted in a journal focused on the interests of oncology.

We take note of this Reviewer’s comment. However, we are not convinced that is the case. The purpose of this work is to corroborate previous findings in a uniform and then more correct cell context. To this end we repeated twice the experiments using a clone of HCT116 cells expressing p14ARF ectopically and the results perfectly confirmed what we already published on a population of HCT116 cells re-expressing p14ARF. Therefore, we consider these results sufficient to conclude that p14ARF is involved in the maintenance of cell ploidy. Besides, we believe our study, being basic research, is in line with Genes policy and it perfectly suits with the Special Issue about p14ARF roles.

Major concerns:

  1. In this work, the authors used the single clone of HCT116 cells with tet-on system-mediated p14ARFexpression to performed all experiments. The clonal effect on these phenotypes should be evaluated.

We agree with this Reviewer that sometimes the analysis of just a single clone could be reductive. Our results obtained by analysing this HCT116 clone are very close to the ones obtained previously with a population of HCT116 cells where p14ARF was expressed by a tet-off inducible system both in transient and stably transfected cells (Veneziano et al. 2018). Therefore, we do not think that the observed results are due to a “clonal effect” but rather to a specific effect of p14ARF re-expression.

  1. In Figure 2B, they scored the chromosomal instability using the criteria of <2n, 2n, and >2n. However, this representation is not general. They should show the number or percentage of aneuploid cells in each chromosome number. In general, aneuploidy followed by the spindle assembly checkpoint disruption accompanies both loss and gain of chromosomes. In this manuscript, they showed only hypo-diploidy after the CENP-E inhibitor treatment. To exclude the possibility of artifacts in the preparation of chromosome slides, images of chromosomes in each sample should be shown.

We thank this Reviewer for his/her suggestion. We decided to graph the chromosome number as <2n, 2n and >2n because we thought that in this way the result is clearer. Actually, the suggested kind of graph can be more exhaustive. Therefore, we have followed the suggestion changing the graph and adding representative images of metaphases.

  1. HCT116 cells are immortalized cells. It is too difficult to analyze the cellular senescence in the HCT116 cells. Although SA-b-Gal is indeed a senescence marker, it is too risky to conclude the insists based on only Figure 2D. Thus, I recommend to delete the data and descriptions of cellular senescence.

We agree with the Reviewer that analysing cell senescence in immortalized cells like HCT116 is not straightforward. However, these cells are usually used in several lab to test drugs inducing senescence and several reports are present in the scientific literature (i.e Wang et al.Cell Death and Disease 2018; Bojko et al. Cells 2019; Kitada et al. Oncology Letters 2019; Mosieniak et al. Mechanisms of Ageing and Development 2012).

Anyway, wishing to follow the Reviewer’s suggestion we have deleted the graph about SA-beta-gal staining in the revised version of the manuscript considering that it does not add important information to our study.

Minor concerns

  1. The error bars of all graphs are standard deviations or standard errors? They should clarify this point.

We thank the Reviewer for pointing out this aspect. In all the graphs the standard error of the mean is indicated. We have added this information in the legend of the figures.

  1. They did not perform any statistical analyses.

Although the experiments were done twice we observed that, as discussed above, these two replicates perfectly confirmed what we already published on a population of HCT116 cells ectopically expressing p14ARF, so we think that additional replicates would not be necessary to draw the conclusions.    

Reviewer 3 Report

The Short Communication is very well structured and updates on recent new role of p14ARF  in the maintenance of chromosome stability.

The following minor corrections should be taken into account:

  1. In the introductory section, this paragraph should be reviewed and supplemented, mentioning the consequences of the alterations due to loss of function of CDKN2a and the current technologies that allow them to be identified, such as NGS, aCGH, MLPA, etc:

“Considering that p14ARF  is a tumor suppressor it is not surprising that most human cancers colorectal, gastric, prostate, breast bladder, oral, hepatocellular, lung cancers, glioblastoma and belanoma) are characterized by its inactivation mainly due to promoter hypermethylation, homozygous deletions and mutations (reviewed in[1]) or just reduced levels whose cause has not been elucidated yet[7]”

It would be interesting if the authors discuss the following aspects.:

  1. It would be interesting to argue that if p14ARF triggers the apoptotic pathway to halt aneuploid cell proliferation, how this could contribute to the regulation of chromosomal stability. Are there cell or animal models about it?. 
  2. Are there studies of chromosomal instability in cell models of cancer? What results have yielded?
  3. Are there studies of chromosomal instability in patients with alterations in the CDKN2a locus.

Author Response

Responses to Reviewers’ comments:

(in blue and italics the Authors’ responses).

Reviewer 3

The Short Communication is very well structured and updates on recent new role of p14ARF  in the maintenance of chromosome stability.

We thank this Reviewer for the positive comment on our work.

The following minor corrections should be taken into account:

  1. In the introductory section, this paragraph should be reviewed and supplemented, mentioning the consequences of the alterations due to loss of function of CDKN2a and the current technologies that allow them to be identified, such as NGS, aCGH, MLPA, etc:

“Considering that p14ARF  is a tumor suppressor it is not surprising that most human cancers colorectal, gastric, prostate, breast bladder, oral, hepatocellular, lung cancers, glioblastoma and belanoma) are characterized by its inactivation mainly due to promoter hypermethylation, homozygous deletions and mutations (reviewed in[1]) or just reduced levels whose cause has not been elucidated yet[7]”

We thank this Reviewer for the suggestion and we have added the following paragraph in the revised text:

“The importance of the role of p14ARF is highlighted by the effect of its loss in mouse models: mice were highly prone to develop tumors and mostly died during their first year of life. In addition, p14ARF-null mouse embryonic fibroblasts can overcome the senescence crisis and proliferate continuously, and they also can be easily transformed by the oncogene Ras [7].“

……….

“Due to its frequent alteration in cancer, population studies have been published to show with the most recent genome-wide approaches, namely Next Generation Sequencing and array-CGH, the most recurrent p14ARF alterations in melanoma and leukaemia”

It would be interesting if the authors discuss the following aspects.:

  1. It would be interesting to argue that if p14ARF triggers the apoptotic pathway to halt aneuploid cell proliferation, how this could contribute to the regulation of chromosomal stability. Are there cell or animal models about it?. 

We are not sure to have correctly understood this Reviewer’s comment. We show that p14ARF presence is necessary to stem aneuploid cell proliferation likely by committing them to apoptosis. Through this mechanism numerical chromosomal stability would be preserved in normal cells. To our knowledge there is no animal model showing this mechanism in adult tissues, while for example mouse models have revealed a lineage-specific depletion of aneuploid cells during development (Bolton et al. Nature Communications 2016).

  1. Are there studies of chromosomal instability in cell models of cancer? What results have yielded?

Lots of cell and mouse models of chromosome instability (CIN) have been generated in the last 20 years. The main conclusions are that by itself chromosome instability is not sufficient to cell transformation and it is unfavourable for cell proliferation. However, CIN can facilitate cancer initiation and progression in particular genetic backgrounds. If the Reviewer is interested in this topic we can suggest a nice chapter by Simon et al. published in 2015 by Springer as Part of the Recent Results in Cancer Research book series.

  1. Are there studies of chromosomal instability in patients with alterations in the CDKN2a locus.

To our knowledge there is no study specifically describing chromosome instability in patients with p14ARF mutations. Anyway, it has been reported that patients with mutations in p14ARF have high risk of developing melanoma and pancreatic cancer (McWilliams et al. Eur J Hum Genet. 2011; De Unamuno et al. Melanoma Res. 2018). P14ARF is also inactivated by several mechanisms in a large number of human tumors (Inoue and Fry; Tumor Microenviron 2019).

Round 2

Reviewer 2 Report

In this revised manuscript, the authors addressed all of my concerns. Therefore, I recommend publication of this study in Genes. 

Author Response

We thank this Reviewer for the positive comment on our revised manuscript and for his/her recommendation to publish it in Genes.

This manuscript is a resubmission of an earlier submission. The following is a list of the peer review reports and author responses from that submission.

Round 1

Reviewer 1 Report

The manuscript entitled “P14 ARF: the absence that makes the difference” is a short communications that shows how p14ARF have an impact in eliminating aneuploid cells in order to keep chromosome balance.

Data here presented is not new at all since the same authors already have published more comprehensive and abundant similar data in previous reports (https://doi.org/10.1002/jcp.24976; https://doi.org/10.1007/s00438-018-1495-5) showing that indeed p14ARF can function as a aneuploidy suppressor by eliminating cells that have undergone chromosome miss-segregation by different means.

The current submitted manuscript, does not add any new relevant data, and only repeats what authors have already published recently.

In addition, the manuscript lacks some important evidences:

1.- authors claim that p14ARF overexpression does not alter cell proliferation, but this experimental evidence is not shown. Since all subsequent data is related to cell survival, I find critical to show the cell proliferation experimental data set.  Cell cycle profiling by FACS is not enough to conclude that cells proliferate at the same rate.

2.- regarding the analysis of aneuploidy, this is done by mitotic chromosome spreads. It surprise me that the effect of GSK923295, even though is at a very low concentration (5nM), does not lead to an increase in G2/M populations as published by many others (for example: doi: 10.18632/oncotarget.4879). I would expect to have an increase in tetraploid or polyploidy cells rather hypodiploid.

3.- Would be interesting to have a more detailed study of the cells mitotic fate when treated with GSK923295+DOX. Count mitotic aberrations, (lagging chromosomes, chromosome bridges, micronuclei, Spindle shape, etc). This can give interesting data of how p14ARF can deal with mitotic aberrancies.

4.- CENP-E inhibition is a very efficient way to generate aneuploidy. To make this data more robust would be interesting to evaluate if p14ARF expression can compensate aneuploidy generated by other means (Mps1 inhibition, Eg5 inhibition-recovery, low concentrations of nocodazole).